# The Effects of Potassium Fertilizer on the Active Constituents and Metabolites of Bulbs from *Lilium davidii* var. *unicolor*

**Lei Jin [1,†], Qing Yuan [2], Jiao Bi [3,†], Gang Zhang [2,\*] and Ping Zhang [1,\*]**

[1]  College of Forestry and Prataculture, Ningxia University, Yinchuan 750021, China; jinlei2013@nxu.edu.cn
[2]  College of Landscape Architecture and Arts, Northwest A&F University, Xianyang 712100, China; yuanqing555@nwafu.edu.cn
[3]  Xinxiang Institute of Engineering, Xinxiang 453000, China; bi_jiao@163.com
[\*]  Correspondence: zhanggang@nwafu.edu.cn (G.Z.); zp2009@mail.bnu.edu.cn (P.Z.)
[†]  These authors contributed equally to this work.

**Abstract:** *Lilium davidii* var. *unicolor* (Lanzhou lily) is rich in nutrients, making it an important economic plant widely used in the fields of food and medicine. In this study, potted lily bulbs were treated with nutrient solutions containing K+ and nutrient solutions without K+ (CK and KT). The contents of nutrients in lily bulbs at different stages after treatment were compared. It was found that the application of potassium fertilizers increased the content of total phenols, flavonoids, and flavanols in lily bulbs and the antioxidant activity in the bulbs. Simultaneously, the study observed that potassium fertilizers could impact the accumulation of polysaccharides and saponins. Furthermore, employing non-targeted metabolomics, the secondary metabolites of mature Lanzhou lily bulbs were scrutinized both with and without potassium fertilization (KT and CK). A total of 607 metabolites were identified, including 573 in positive ion mode and 34 in negative ion mode. These metabolites were classified into 13 categories at the superclass level, with lipids and lipid molecules (37.93%), organic acids and their derivatives (16.52%), organic oxygen compounds (14.88%), and phenylpropanoids and polyketides (13.61%) being the most prominent. Differential metabolite enrichment analysis between the experimental and control groups showed that the differential metabolites were mainly concentrated in metabolic pathways related to amino acid biosynthesis, such as arginine and proline metabolism, beta-alanine metabolism, alanine, aspartate, and glutamate metabolism. Additionally, it was found that the application of potassium fertilizer increased the accumulation of amino acids in Lanzhou lily bulbs. Overall, this study provides a theoretical reference for the development of nutrients and efficient cultivation techniques for *L. davidii* var. *unicolor* bulbs.

**Keywords:** Lanzhou lily; antioxidant activity; metabolomics; amino acid

## 1. Introduction

*Lilium davidii* var. *unicolor* belongs to the *Lilium* genus of the Liliaceae family and is an important economic crop widely cultivated in Lanzhou, China, earning it the name Lanzhou lily [1]. The high polysaccharide content of the bulb makes it the only type of sweet lily that is edible. The bulbs also contain starch, protein, and dietary fiber, making them a medicinal and edible plant in China [2]. Lanzhou lily is a traditional medicinal plant and a bulb plant consumed by the public, with functions that include clearing heat and toxins, nourishing the lungs, fighting cancer, and enhancing human immunity [3]. It is widely used in food and medicine [4].

The bulb, an important organ of reproductive development, develops from the axillary meristem along the stem of the mother plant. Bulbs usually grow in spring and bloom in summer and autumn. The formation and vigorous development of bulbs are crucial factors for maintaining the normal life of the Lanzhou lily. As the main part responsible for storing nutrients, the bulb plays a significant role in the growth of the Lanzhou lily. Additionally, studies have shown that Lanzhou lily bulbs contain a large number of flavonoids and

phenolic acids that have antioxidant and antibacterial effects [5]. It has been reported that different growth stages can affect the accumulation of nutrients in plant organs, as observed in plants such as Echinacea [6], *Scutellaria baicalensis* [7], and peonies [8].

The accumulation of nutrients in plant organs is known to be influenced by cultivation conditions. Potassium is a crucial element in plant growth and is the most important and abundant nutrient ion found in living plant cells. It plays a significant role in numerous physiological and biochemical processes, such as protein synthesis, photosynthesis, ion homeostasis, enzyme activation, stomatal movement, and osmotic pressure regulation [9]. During the cultivation process of horticultural plants, fertilizers are used to ensure the plants' normal growth and improve their quality. Potassium fertilizer has been found to affect the accumulation of nutrients in plants [10]. Several studies have shown that the application of potassium fertilizer can increase the content of polysaccharides and other macromolecules in Lanzhou lily bulbs. Moreover, potassium is essential for plant osmotic pressure regulation and phloem transport, and some enzymes in plants require potassium for activation. In various plants such as alfalfa [11], sugar beet [12], cotton [13], and soybean [14], potassium deficiency conditions lead to reduced primary metabolite content. Additionally, studies indicate that potassium deficiency leads to a reduced accumulation of phenolic acids [15] and flavonoids [16]. These findings highlight the significance of potassium in the production and accumulation of plant metabolites.

Previous research efforts have primarily centered around the detection and quantification of macromolecular compounds present in Lanzhou lily bulbs. However, the present study employs a metabolomics approach to examine the composition of secondary metabolites within Lanzhou lily bulbs across varying stages of growth and subsequently assesses the influence of potassium fertilizer application on their accumulation. This study will provide a reference for the determination of nutrient components and the cultivation and fertilization management strategies of Lanzhou lily bulbs.

## 2. Materials and Methods

### 2.1. Plant Materials

Test material, including Lanzhou lily bulbs, 12–15 cm in circumference, was purchased from Huamu Company, Lanzhou City, Gansu Province, China. These bulbs had a circumference of 12–15 cm. The experiment was conducted in a glass greenhouse situated at Northwest A&F University in China, where the average indoor temperature was maintained at 24 °C and natural light was provided for over 10 h per day. Moreover, the relative humidity of the greenhouse was kept between 70% and 80%.

### 2.2. Experimental Design

In this experiment, two treatments were carried out: CK, the nutrient solution containing no K+, and KT, the 2× Hoagland nutrient solution (the concentrations after the balance are shown in Table 1). A total of 30 lily bulbs were processed for each treatment method, for 60 lily bulbs. A plastic flowerpot measuring 20 cm × 20 cm was chosen, and 3 kg of substrate (comprising peat and perlite at a ratio of 2:1) was placed in each pot for the lilies. One corm was planted in each pot, with a depth of approximately 10 cm. Each treatment method applied the above two solutions every 5 days, for a total of 3 times. Lily bulbs were collected 30, 45, 60, 75, 90, 105, and 120 days after the last application of the above solution. Four lily bulbs were randomly collected from each collection, mixed together, washed, and placed at −80 °C for experiments.

**Table 1.** Different potassium treatments were applied to the seedlings of *L. davidii* var. *unicolor* (mg/L).

| Treatment | CK | K3 |
|---|---|---|
| N (609.8 mg/L) and P (88.4 mg/L) | 0 | 895.2 |

### 2.3. Sample Extraction

In this experiment, 5 g of the sample was extracted using a solution consisting of 100 mL of methanol/water (4:1, *v*/*v*). The mixture was allowed to settle at a temperature of −10 °C and treated using a High-Throughput Tissue Crusher Wonbio-96c (Shanghai Wanbo Biotechnology Co., Ltd., Shanghai, China) at a frequency of 50 Hz for 8 min, followed by ultrasound at 40 kHz for 25 min at 4 °C. Subsequently, the samples were placed at a temperature of −20 °C for 25 min to precipitate proteins. After centrifugation at 13,000× *g* for 15 min, the supernatant was carefully transferred to sample vials and stored at −80 °C for future analysis.

### 2.4. Determination of Total Phenolics, Flavonoids, and Flavanol Contents

Total phenolic content, flavonoid content, and flavanol content were measured using the reported protocol [15–17].

### 2.5. Determination of Saponins and Polysaccharide Contents

The saponin content and polysaccharide content were determined by the vanillin-perchloric acid method [18,19].

### 2.6. Untargeted Metabolomics Analysis

#### 2.6.1. LC–MS Analysis

A 2 µL sample was separated by an HSS T3 column (100 mm × 2.1 mm, 1.8 µm) and subsequently entered mass spectrometry detection. The mobile phase consisted of 0.1% formic acid in water/acetonitrile (95:5, *v*/*v*) (solvent A) and 0.1% formic acid in acetonitrile/isopropanol/water (47.5:47.5:5, *v*/*v*) (solvent B). The solvent gradient changed according to the following conditions: From 0 to 0.1 min, the proportion of solvent B changed from 0% to 5%; from 0.1 to 2 min, the proportion of solvent B changed from 5% to 25%; from 2 to 9 min, the proportion of solvent B changed from 25% to 100%; from 9 to 13 min, the proportion of solvent B was maintained at 100%; from 13 to 13.1 min, the proportion of solvent B changed from 100% to 0%; and from 13.1 to 16 min, the proportion of solvent B was maintained at 0% for equilibration of the system. The sample injection volume was 5 µL, and the flow rate was set at 0.3 mL/min. The column temperature was kept constant at 30 °C.

The mass spectrometric data were equipped with an electrospray ionization (ESI) source operating in either positive or negative ion mode (Thermo UHPLC-Q Exactive, Shanghai, China). The optimal conditions for ESI were set as follows: heater temperature, 400 °C; capillary temperature, 320 °C; ion spray voltage floating (ISVF), 2800 V in negative mode and 3500 V in positive mode, respectively; and normalized collision energy, 20–40–60 V rolling for MS/MS. Full MS resolution was set at 70,000, and MS/MS resolution was set at 17,500. Data acquisition was performed with the Data Dependent Acquisition (DDA) mode, and the detection was carried out over a mass range of 70–1050 *m/z*.

#### 2.6.2. Data Processing

The LC/MS raw data obtained from the experiment was first preprocessed by the Progenesis QI v3.0 software, developed by Waters Corporation in Milford, MA, USA. The resulting data was uploaded to MetaboAnalyst 5.0, a web-based software platform "https://www.metaboanalyst.ca/" (accessed on 12 May 2023) that provides a comprehensive set of tools for metabolomic data analysis. On this platform, the data was processed, including baseline filtering, peak identification, integration, retention time correction, peak alignment, and normalization, followed by pattern recognition. Additionally, metabolites were searched and identified with the HMDB "http://www.hmdb.ca/" (accessed on 12 May 2023), Metlin "https://metlin.scripps.edu/" (accessed on 12 May 2023), and Massbank "https://massbank.eu/MassBank/" (accessed on 12 May 2023) databases. Only metabolic signatures detected in at least 80% of the samples were retained after filtering, and minimum metabolite values were imputed for specific samples in which the metabolite levels fell below the lower limit of quantitation. To minimize errors arising from sample prepara-

tion and instrument instability, the response intensity of the sample mass spectrum peaks was normalized by the sum normalization method, resulting in a normalized data matrix.

### 2.7. Determination of Antioxidant Activity

The DPPH free radical scavenging assay is a hydrogen atom transfer (HAT) reaction that is widely used to determine the antioxidant capacity of biological materials [20]. The DPPH radical scavenging activity assay was performed as described by Taslimi [21]. The cupric-ion-reducing capacity (CUPRAC) was determined as described by Apak, Guculu, Ozyurek, and Karademir [22].

### 2.8. Statistical Analysis

All analyses were performed in triplicate, and the results, expressed as mean $\pm$ standard deviation (SD), were analyzed using SPSS version 16.0. A Student's *t*-test was applied to determine the statistical significance of the data.

For non-targeted metabolomics data, after data preprocessing, variance analysis was conducted on the resulting matrix file. The R package ropls (Version 1.6.2) were used to perform principal component analysis (PCA) and partial least squares discriminant analysis (PLS-DA). The selection of significantly different metabolites was based on the variable importance in the projection (VIP) obtained from the PLS-DA model and the *p*-value of the Student's *t*-test, whereby metabolites with VIP > 1 and $p < 0.05$ were considered significantly different. A total of 607 metabolites were screened. Differential metabolites between the two groups were summarized and mapped to their corresponding biochemical pathways through metabolic enrichment and pathway analysis using a database search, namely KEGG "http://www.genome.jp/kegg/" (accessed on 12 May 2023). These metabolites were classified according to the pathways in which they were involved or the functions they performed. Enrichment analysis was carried out to determine whether a function node was enriched with a group of metabolites, whereby the annotation analysis of a single metabolite developed into an annotation analysis of a group of metabolites. Fisher's exact test was used to identify statistically significantly enriched pathways, utilizing the Python package SciPy "https://docs.scipy.org/doc/scipy/" (accessed on 12 May 2023).

### 3. Results

#### 3.1. Total Phenolic, Flavonoid, and Flavanol Content

In CK and KT, the total phenolic content of lily bulbs ranged between 0.799–0.934 mg/g and 0.805–1.132 mg/g, respectively. It showed an increasing trend after 30–120 days under different treatment conditions (Figure 1A). Notably, the introduction of potassium fertilizer yielded an appreciable enhancement in the overall phenolic content of lily bulbs compared to the CK group. Simultaneously, the content of total flavonoids in lily bulbs (0.056–0.095 mg/g in CK and 0.022–0.159 mg/g in KT) also showed a similar trend as total phenols. The content of total flavonoids in lily bulbs could also be increased after potassium fertilizer application (Figure 1B). Conversely, 30–120 days after fertilization, the content of total flavanols in lily bulbs (0.083–0.014 mg/g) showed a decreasing trend (Figure 1C). Remarkably, lily bulbs have matured 120 days after fertilization, which is the harvest period for lily bulbs. Interestingly, compared with the lily bulbs at the harvest stage, the total phenolics, total flavonoids, and total flavanol contents of the lily bulbs treated with potassium fertilizer were significantly higher than those of the CK group.

#### 3.2. Polysaccharide and Saponin Content

Notably, the application of potassium fertilization has been found to engender an elevation in the saponin content within lily bulbs. Among them, under different treatment time conditions, saponin content ranged from 56.55 to 597.37 mg/g for KT and from 70.99 to 692.96 mg/g for CK, while polysaccharide content ranged from 844.93 to 192.18 mg/g for KT and from 598.98 to 330.61 mg/g for CK. As shown in Figure 2A, the saponin content in lily bulbs increased 30–120 days under different treatment conditions, and the CK also

showed the same trend. On the contrary, with the increase in growth time, the content of polysaccharides in lily bulbs decreased gradually (Figure 2B). Nevertheless, at the harvest stage (120 days after fertilization), the saponin and polysaccharide content in lily bulbs applied with potassium fertilizer were significantly higher than those in the CK group.

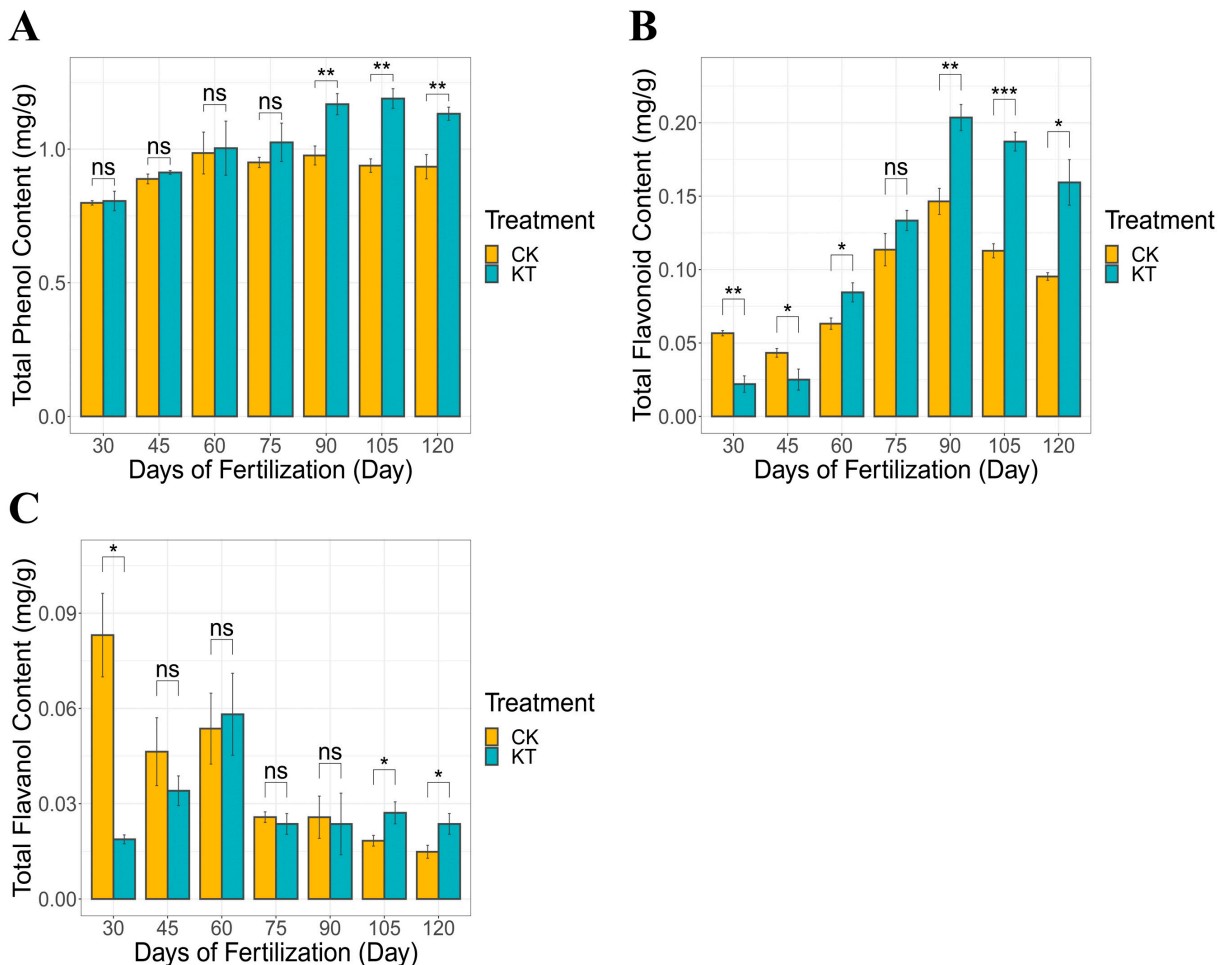

**Figure 1.** Contents of total phenols, flavonoids, and flavanols in bulbs of different treatments ((**A**) total phenol; (**B**) total flavonoid; (**C**) total flavanol). The data are shown as mean ± S.D. (n = 3). Significant differences between treatments were determined by the Student's *t*-test: * $p < 0.05$; ** $p < 0.01$, *** $p < 0.01$, ns meant no signifacance.

### 3.3. Antioxidant Activity

The DPPH scavenging ability exhibited a range of 834.26 to 4489.66 μmol/g for KT and 720.42 to 4289.32 μmol/g for CK (Figure 3A). The application of potassium fertilizer resulted in an enhancement of DPPH scavenging capability relative to CK. Concurrently, the copper ion reducing capacity displayed values spanning 1095.82 to 1297.75 μmol/g for KT and 1118.32 to 1192.49 μmol/g for CK. Significantly increased following potassium fertilizer application. Similarly, the copper-ion-reducing ability of lily bulbs following potassium fertilizer application was also higher than in CK.

### 3.4. Metabolite Profiling and Classification

Through the above analysis, it was determined that the utilization of potassium fertilizer exerted a discernible impact on various constituents of lily bulbs, including total phenolic acids, flavonoids, flavanols, saponins, polysaccharides, and antioxidant activities during the harvest period. Afterwards, secondary metabolites were identified in lily bulbs harvested at maturity (120 days after fertilization) using non-targeted metabolomics.

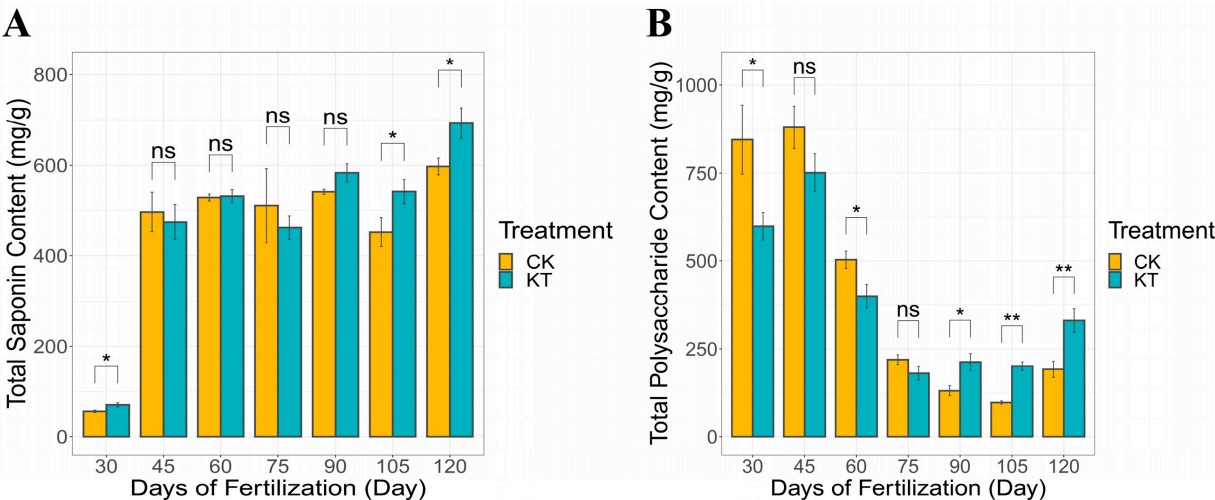

**Figure 2.** Contents of saponins and polysaccharide content in bulbs of different treatments ((**A**) saponin content; (**B**) polysaccharide content). The data are shown as mean ± S.D. (n = 3). Significant differences between treatments were determined by the Student's *t*-test: * $p < 0.05$; ** $p < 0.01$, ns meant no significance.

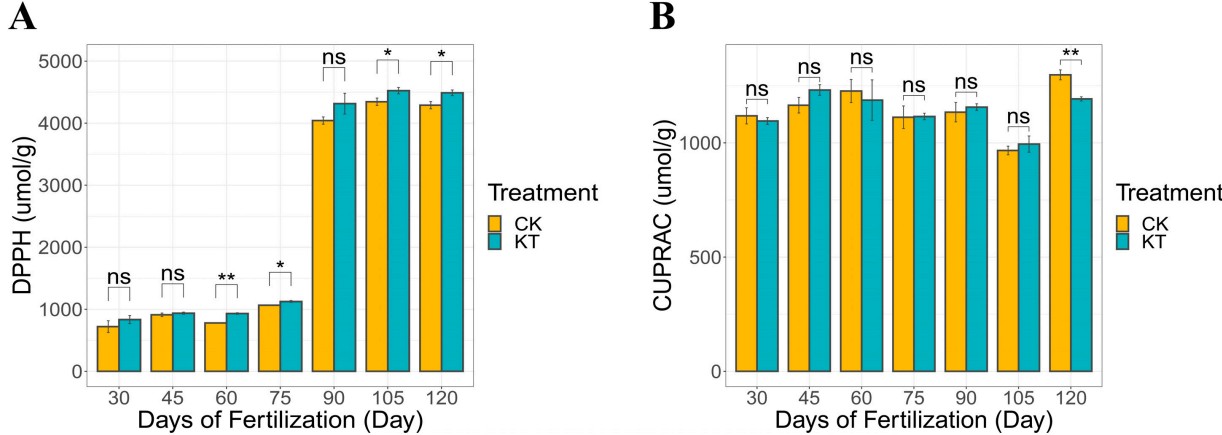

**Figure 3.** Antioxidant activity was determined by the DPPH and CUPRAC assays ((**A**) DPPH assays; (**B**) CUPRAC assays). The data are shown as mean ± S.D. (n = 3). Significant differences between treatments were determined by the Student's *t*-test: * $p < 0.05$; ** $p < 0.01$, ns meant no significance.

Firstly, the non-targeted metabolomic data were preprocessed. The metabolite peaks detected in different samples were compared using PCA. The total variation in the positive ion mode was explained by PC1 and PC2, which accounted for 63.155% and 17.552%, respectively (Figure 4A). Similarly, in the negative ion mode, PC1 and PC2 explained 55.515% and 22.401% of the total variation, respectively (Figure 4B). The results of the PCA analysis showed that CK and KT were clearly separated. To further examine the reliability of the metabolomics data, PLS-DA was conducted at the metabolite accumulation level to establish a relationship model for distinguishing sample groups. Since the PLS-DA method grouped the samples when building the model, the results may have the problem of overfitting. Permutation plots can effectively evaluate whether the current model is overfitted (Figure S1). The results of the PLS-DA analysis were similar to those of PCA. The two groups of samples could be separated, which indicated that potassium application had an effect on the metabolites of lily bulbs (Figure 4C,D).

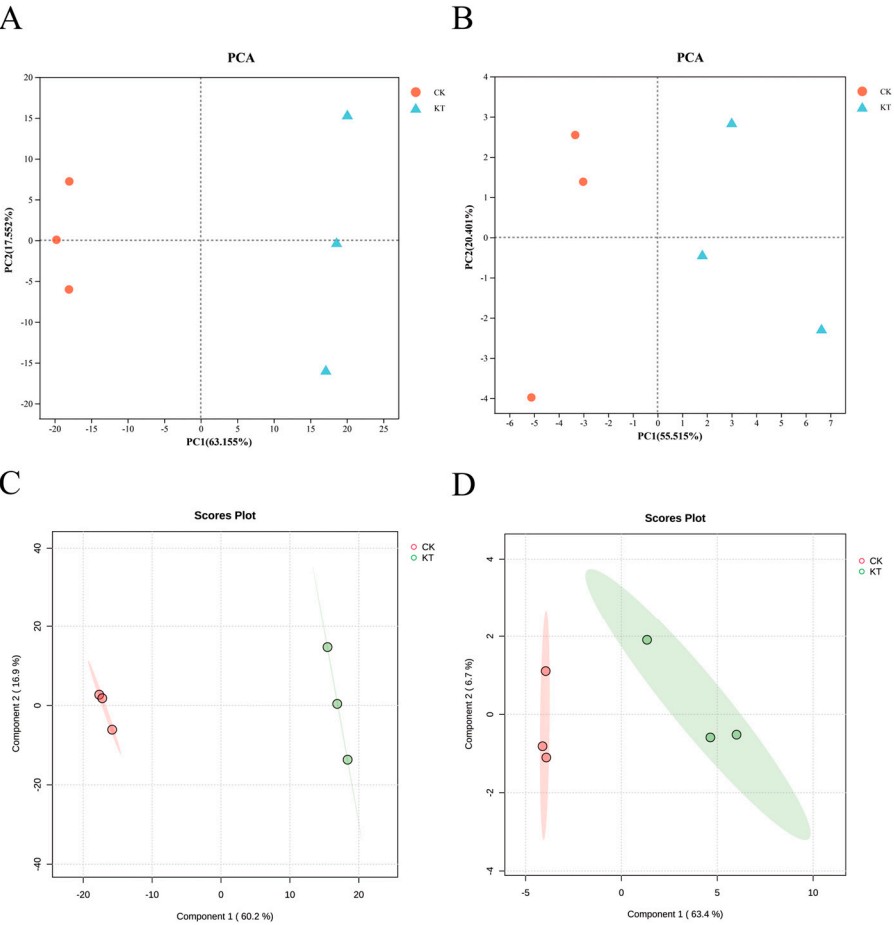

**Figure 4.** Principal component analysis (PCA) and partial least squares discriminant analysis (PLS-DA) of the Lanzhou lily bulb metabolite of different samples in the positive and negative ion modes ((**A**,**C**) pos.; (**B**,**D**) neg.).

By performing non-targeted LC-MS analysis, a total of 10,936 peaks were detected in different samples. Among these, 6619 peaks were detected in the negative ion mode, and 4317 peaks were detected in the positive ion mode. Ion peaks with inaccurate qualitative results were removed based on the reference threshold. A total of 607 metabolites were detected, including 573 and 34 metabolites in the positive (Figure S2) and negative modes (Figure S3), respectively, as shown in Figure 5A. These metabolites were divided into 13 categories at the superclass level, including lipids and lipid-like molecules (37.93%), organic acids and derivatives (16.52%), organic oxygen compounds (14.88%), phenylpropanoids and polyketides (13.61%), organ heterocyclic compounds (8.17%), and benzenoids (3.45%), as shown in Figure 5B.

### 3.5. Differentially Accumulated Metabolite (DAM) Identification and Analysis

To investigate the distribution of metabolites in different samples, further analysis was performed according to the abundance of metabolites in different samples. Overall, the results suggest that metabolic profiles differ between samples. To study the metabolic characteristics among different samples, differential abundance analysis was further conducted according to the screening criteria of VIP > 1 and *p*-value < 0.05 by *t*-test. As shown in Figure 6A, a total of 129 differential metabolites were obtained in the two groups of samples, among which 59 metabolites were significantly up-regulated and 70 metabolites were significantly down-regulated in CK compared with KT (Table S1). This phenomenon indicated that the content of metabolites in KT was increased compared with CK, which also reflected that potassium application could affect the accumulation of metabolites in lily bulbs.

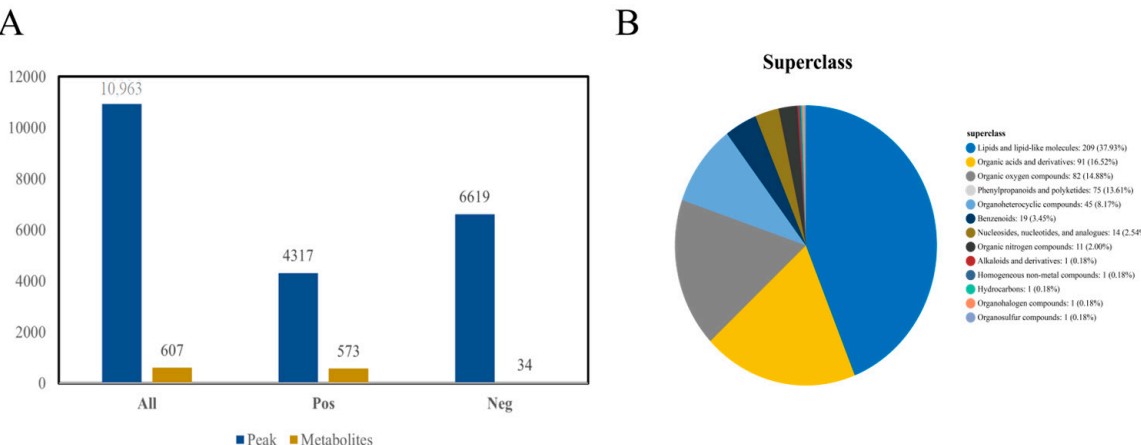

**Figure 5.** Classification statistics of the identified metabolites from Lanzhou lily bulbs. (**A**) Statistical diagram of the substance peaks and metabolites. (**B**) Classification ring diagram of the identified metabolites at the superclass level.

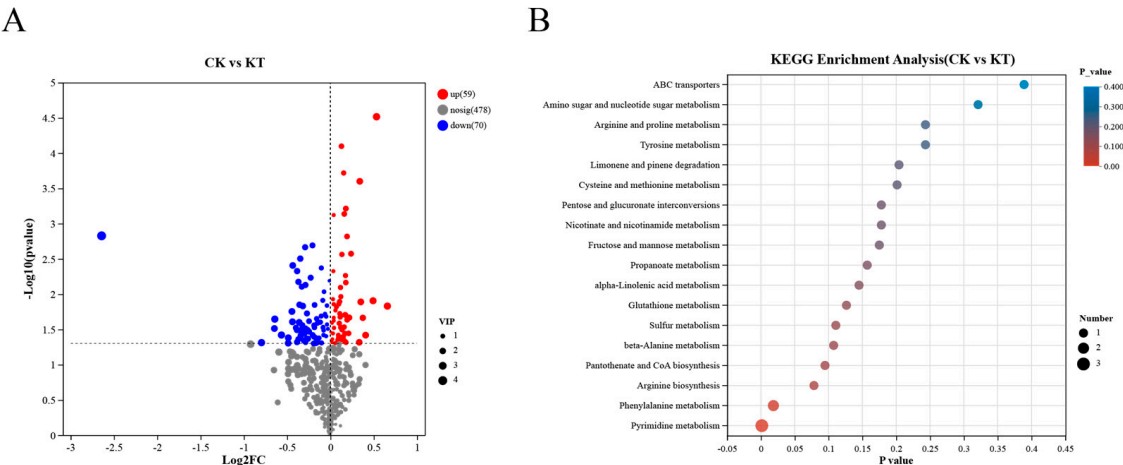

**Figure 6.** Volcano diagram and KEGG enrichment analysis of DAMs identified from the different samples. (**A**) Volcano diagram of CK vs. KT; (**B**) KEGG enrichment analysis of CK vs. KT.

To investigate the functional enrichment of differential metabolites among different samples, a functional enrichment analysis was conducted using the KEGG database. DAMs were enriched in many pathways in different experimental groups. Specifically, 18 pathways were enriched in the comparison of CK vs. KT, respectively (Figure 6B). Most of the differentially accumulated metabolites were annotated in the amino acid synthesis pathway, including arginine and proline metabolism, beta-alanine metabolism, alanine, aspartate, and glutamate metabolism. These results suggest that potassium application can affect the above metabolites and metabolic pathways.

## 4. Discussion

The bulbs of Lanzhou lily have been a subject of interest among researchers due to their rich nutrient content [23–25]. With the expansion of the Lanzhou lily planting area, efficient cultivation methods have become the focus of attention. As one of the important tasks in horticultural production, fertilization will affect the growth and production of horticultural crops to a large extent [26]. Nitrogen (N), phosphorus (P), and potassium (K) are the main mineral elements that play an important role in plant growth [27].

Among these elements, potassium (K) is beneficial for plants to obtain higher photosynthesis and affects the growth and development of plants. It plays a vital role in various plant developmental processes, including enzyme activation, photosynthesis, carbohydrate

metabolism, and abiotic stress response [28,29]. The study found that the total phenolic and total flavonoid contents of lily bulbs after potassium fertilization (KT) were higher than those of the non-potassium fertilization group (CK). Studies have pointed out that potassium fertilizer can affect the content of total phenols and flavonoids in plants [30]. Moreover, plants are rich in phenolic acids and flavonoids, which also make them a raw material for natural antioxidants. It was found that the antioxidant activity of lily bulb extract was also affected by the increase in total phenolic and flavonoid content.

The bulbs of Lanzhou lily contain polysaccharides and saponins, making them edible [31]. The accumulation of nutrients in plants is related to the environment, growth period, fertilization, and other conditions [32]. This study found that in lily bulbs, the polysaccharide content gradually decreased with the growth period. It was also found in sea buckthorn that the polysaccharide content decreased with the increase in growth period, and the content changes in different tissues were different [33]. On the contrary, the saponin content in lily bulbs gradually increases at different stages. Meanwhile, the polysaccharide and saponin contents in KT were higher than those in CK in Lanzhou lily bulbs obtained at harvest time. This also indicated that potassium fertilizer could increase the accumulation of polysaccharides and saponins in lily bulbs. Fertilization is an important process in the production of horticultural crops, with previous studies indicating that it can increase the content of polysaccharides and saponins in fruits [18,34].

At the same time, plants are rich in secondary metabolites, which play an important role in plant growth and development, and secondary metabolites also have active effects, so they are used as raw materials and widely used in the field of medicine [35]. The accumulation and formation of secondary metabolites are affected by plant growth conditions and developmental stages [36,37]. While non-targeted metabolomics is commonly used in plant studies, it was applied in this study to investigate the bulbs of the Lanzhou lily. Through comparative analysis, 607 metabolites (573 positive ions and 34 negative ions) were identified, and differences in metabolite content were observed among bulbs. Differential analysis of metabolites revealed 202 differential metabolites, which were primarily enriched in pathways such as arginine and proline metabolism, beta-alanine metabolism, and alanine, aspartate, and glutamate metabolism. These pathways are associated with amino acids, which play an important role in the growth and development of plants and the transformation of nutrients [38]. This also indicated that potassium fertilizer could affect the accumulation of secondary metabolites in lily bulbs.

## 5. Conclusions

In this study, the contents of total phenols, flavonoids, and flavanols in lily bulbs with and without potassium fertilizer application at different growth stages were compared. It can be found that potassium fertilizer increased the content of total phenols, flavonoids, and flavanols in lily bulbs, enhancing the antioxidant activity (DPPH and CURRPC) of lily bulbs. On the other hand, the application of potassium fertilizer increased the contents of polysaccharides and saponins in lily bulbs. Using non-targeted metabolomics, the secondary metabolites of lily bulbs after different treatments were measured. A total of 607 metabolites were detected and identified in both positive and negative modes. Enrichment analysis of differential metabolites in different experimental groups revealed that the application of potassium fertilizer led to an increase in the amino acid content. Therefore, this study has significant theoretical implications for the development of nutrient-rich Lanzhou lily bulbs and the advancement of cultivation techniques.

**Supplementary Materials:** The following supporting information can be downloaded at: https://www.mdpi.com/article/10.3390/horticulturae9111216/s1, Figure S1: All metabolites identified in positive ion mode of the Lanzhou lily bulbs between CK and KT; Figure S2: All metabolites identified in negative ion mode of the Lanzhou lily bulbs between CK and KT; Figure S3: All metabolites identified in negative ion mode of the Lanzhou lily bulbs between CK and KT.

**Author Contributions:** Conceptualization, P.Z. and G.Z.; methodology, L.J. and Q.Y.; software, J.B.; resources, L.J. and Q.Y.; data curation, Q.Y.; writing original draft preparation, L.J., Q.Y. and J.B.; writing, reviewing, and editing, P.Z. and G.Z.; supervision, P.Z. and G.Z. All authors have read and agreed to the published version of the manuscript.

**Funding:** The present work was financially supported by the Key Research and Development Projects of the Ningxia Hui Autonomous Region (2021BBF02025), the Chunhui Project of the Ministry of Education of China (Z2016028), and First-class Construction Discipline in Western China (Horticul-tureNXYLXK2017B03).

**Data Availability Statement:** Available in section "MDPI Research Data Policies" at https://www.mdpi.com/ethics.

**Conflicts of Interest:** The authors declare no conflict of interest.

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
