# Peer review of "The Effects of Potassium Fertilizer on the Active Constituents and Metabolites of Bulbs from Lilium davidii var. unicolor"

_horticulturae, doi:10.3390/horticulturae9111216_

Round 1

Reviewer 1 Report

Comments and Suggestions for Authors

I read with interest the manuscript entitled Impact of Potassium Fertilizer on the Active Constituents and Metabolites of Lilium davidii var. unicolor Lily Bulbs, in which we sought to evaluate the a metabolomics approach to examine the composition of secondary metabolites within Lanzhou lily bulbs across varying stages of growth, and subsequently assesses the influence of potassium fertilizer application on their accumulation. The subject of the article is important and has great relevance for the scientific environment of the study area. Therefore, the manuscript needs some adjustments so that it can then be forwarded to the publication process. The manuscript has the potential for publication in the journal Horticulturae and needs the following adjustments:

TITLE

- Delete the word “impact”.

ABSTRACT

- This section starts with a short introduction and then the results and conclusion are described.

- It is necessary to add information about the objectives, material, and methods, and after that talk about the main results.

- This section needs to be rewritten according to the suggestions above.

- Keywords repeated in the title should be replaced by other words related to the research topic.

INTRODUCTION

- Add references to several important information present in the first paragraph and in the others present in the Introduction.

- In the last paragraph I did not understand why it was mentioned that this study will provide a theoretical reference.

MATERIAL AND METHODS

- The experimental design generated doubts. What design was used? Were 30 repetitions used for each treatment? Was it completely randomized or in blocks? This information needs to be more detailed.

RESULTS

- In the Results, it is necessary to describe the results and then cite the Figure. The authors should reorganize this way of presenting the results.

- Was there no comparison between the fertilizer application times?

- The titles of the axes of the PCA figures must be standardized.

DISCUSSION

- The discussion is written correctly, however, more information is needed to be discussed with other research related to the theme of this study.

CONCLUSION

- The conclusions need to be clear and objective. It needs to be reduced. As it stands, it appears to be part of the Results. Try to reduce these conclusions as much as possible.

Author Response

Response to Reviewer 1 Comments

1. Summary

Thank you very much for taking the time to review this manuscript. Your valuable comments are greatly appreciated. I have revised the manuscript based on your feedback.

2. Questions for General Evaluation

Reviewer’s Evaluation

Response and Revisions

Does the introduction provide sufficient background and include all relevant references?

Can be improved

Are all the cited references relevant to the research?

Can be improved

Is the research design appropriate?

Can be improved

Are the methods adequately described?

Can be improved

Are the results clearly presented?

Can be improved

Are the conclusions supported by the results?

Can be improved

3. Point-by-point response to Comments and Suggestions for Authors

Comments 1: TITLE- Delete the word “impact”.

Response 1: Thank you for your suggestion. We agree with your point of view and have revised the question.

Comments 2:

ABSTRACT

- This section starts with a short introduction and then the results and conclusion are described.

- It is necessary to add information about the objectives, material, and methods, and after that talk about the main results.

- This section needs to be rewritten according to the suggestions above.

- Keywords repeated in the title should be replaced by other words related to the research topic.

Response 2: Thank you for your valuable opinions. Based on your opinions, I have improved the summary part, added the sample processing situation, and also changed the keywords.

Comments 3:

INTRODUCTION

- Add references to several important information present in the first paragraph and in the others present in the Introduction.

- In the last paragraph I did not understand why it was mentioned that this study will provide a theoretical reference.

Response 3: I have added references to the existing ideas, and at the same time, considering that this study is more practical, I have readjusted the content in the last paragraph.

Comments 4:

RESULTS

- In the Results, it is necessary to describe the results and then cite the Figure. The authors should reorganize this way of presenting the results.

- Was there no comparison between the fertilizer application times?

- The titles of the axes of the PCA figures must be standardized.

Response 4: I have improved the description of the results. This manuscript does not conduct research on the number of fertilization applications. The specific process of principal component analysis refers to previous literature [1]. The main purpose is to see whether there are differences between samples.

Comments 5:

DISCUSSION

- The discussion is written correctly, however, more information is needed to be discussed with other research related to the theme of this study.

Response 5: Thank you for your suggestions. We have further improved the discussion part of the paper so that it focuses on the core content of the article.

Comments 6:

CONCLUSION

- The conclusions need to be clear and objective. It needs to be reduced. As it stands, it appears to be part of the Results. Try to reduce these conclusions as much as possible.

Response 6: We have streamlined the conclusion section and retained the important content.

Reviewer 2 Report

Comments and Suggestions for Authors

Presented research has interesting results, however quality needs to be improved. Please see my comments below:

-Information in methodology is very messy, it needs to be precise, without too much details

-line 183: is 760nm correct??

-It will be good if statistical data presented as a table, where possible to see n numbers, p values and interactions

-Is there any specific reason to use different software for traditional anayzis?

-Discussion part is too short. I suggest to combine result and discussion part to make more sense

Comments on the Quality of English Language

Readability is low. Avoid using random complex English words that doesn't match overall language of manuscript. Simplify where needed.

Author Response

Response to Reviewer 2 Comments

1. Summary

Thank you very much for taking the time to review this manuscript. Your valuable comments are greatly appreciated. I have revised the manuscript based on your feedback.

2. Questions for General Evaluation

Reviewer’s Evaluation

Response and Revisions

Does the introduction provide sufficient background and include all relevant references?

Can be improved

Are all the cited references relevant to the research?

Yes

Is the research design appropriate?

Yes

Are the methods adequately described?

Can be improved

Are the results clearly presented?

Can be improved

Are the conclusions supported by the results?

Yes

3. Point-by-point response to Comments and Suggestions for Authors

Comments 1: Information in methodology is very messy, it needs to be precise, without too much details.

Response 1: We have streamlined the methodology and included references to the cited methods to enhance the readability of the article.

Comments 2: line 183: is 760nm correct??

Response 2: We reconfirmed the experimental method and added the literature referenced during the experimental process to the description of the experimental method.

Comments 3: It will be good if statistical data presented as a table, where possible to see n numbers, p values and interactions.

Response 3: Thanks to your suggestion, we have added the number of samples and p-values from the experimental analysis to the analysis of the paper.

Comments 4: Is there any specific reason to use different software for traditional anayzis?

Response 4: Various software tools are employed for data analysis. We have restructured the data analysis section, and in the context of non-targeted metabolomics, we have simplified the language to enhance readability due to the utilization of multiple databases.

Comments 5: Discussion part is too short. I suggest to combine result and discussion part to make more sense

Response 5: Thank you for your suggestion. Upon reviewing papers in the journal "Horticulturae," I observed that the predominant writing pattern separates the description of results and discussions. Accordingly, we have augmented the discussion section to enhance the depth and richness of our paper.

Reviewer 3 Report

Comments and Suggestions for Authors

This study is well conducted, but there are several points to be addressed regarding the statistical analysis:

1) In order to avoid II Type errors, the authors should employ ANOVA or, at least, correct the p value for Bonferroni correction;

2) There is no validation of the PLS model. Moreover, since several variables are not linear with the time, a linear model is not the more suitable one.

3) In lines 81-83, the description of the categories suggests that while CK is pure water, KT not only contains potassium but also other nutrients. Please comment or, if I misunderstood, better clarify the sentence. If there are other nutrients, another data set would be required, i.e. only water and potassium.

Author Response

Response to Reviewer 3 Comments

1. Summary

Thank you very much for taking the time to review this manuscript. Your valuable comments are greatly appreciated. I have revised the manuscript based on your feedback.

2. Questions for General Evaluation

Reviewer’s Evaluation

Response and Revisions

Does the introduction provide sufficient background and include all relevant references?

Yes

Are all the cited references relevant to the research?

Yes

Is the research design appropriate?

Yes

Are the methods adequately described?

Can be improved

Are the results clearly presented?

Can be improved

Are the conclusions supported by the results?

Can be improved

3. Point-by-point response to Comments and Suggestions for Authors

Comments 1: In order to avoid II Type errors, the authors should employ ANOVA or, at least, correct the p value for Bonferroni correction.

Response 1: Thank you for your suggestion. We have supplemented the data analysis methods and p-values in the data analysis..

Comments 2: There is no validation of the PLS model. Moreover, since several variables are not linear with the time, a linear model is not the more suitable one.

Response 2: Thank you for your suggestion. We have validated the PLS model and included the result file in the attached documents. Additionally, we have provided supplementary explanations within the article.

Comments 3: In lines 81-83, the description of the categories suggests that while CK is pure water, KT not only contains potassium but also other nutrients. Please comment or, if I misunderstood, better clarify the sentence. If there are other nutrients, another data set would be required, i.e. only water and potassium.

Response 3: We apologize for any potential confusion caused by our description in the Materials and Methods section. As a result, we have made revisions to certain portions of the Materials and Methods.

Round 2

Reviewer 1 Report

Comments and Suggestions for Authors

The authors made changes to the manuscript. There is just one suggestion:

Increase the font size in all figures. The reader cannot visualize the information in a pleasant way.

Author Response

Dear Reviewer, we have modified the manuscript according to your requirements and increase the font size in all figures. Thank you very much for your patience and careful review of the manuscript.

Reviewer 2 Report

Comments and Suggestions for Authors

No additional comments

Author Response

Dear reviewer, thank you very much for your patience and careful review of the manuscript.

Reviewer 3 Report

Comments and Suggestions for Authors

The authors improved the manuscript, and it can be accepted in the present form.

Author Response

(The authors gave the same response as above.)
